# Performance of Spectral Photon-Counting Coronary CT Angiography and Comparison with Energy-Integrating-Detector CT: Objective Assessment with Model Observer

**DOI:** 10.3390/diagnostics11122376

**Published:** 2021-12-16

**Authors:** David C. Rotzinger, Damien Racine, Fabio Becce, Elias Lahoud, Klaus Erhard, Salim A. Si-Mohamed, Joël Greffier, Anaïs Viry, Loïc Boussel, Reto A. Meuli, Yoad Yagil, Pascal Monnin, Philippe C. Douek

**Affiliations:** 1Department of Diagnostic and Interventional Radiology, Lausanne University Hospital (CHUV), Rue du Bugnon 46, CH 1011 Lausanne, Switzerland; fabio.becce@chuv.ch (F.B.); reto.meuli@chuv.ch (R.A.M.); 2Faculty of Biology and Medicine (FBM), University of Lausanne (UNIL), CH 1015 Lausanne, Switzerland; damien.racine@chuv.ch (D.R.); anais.viry@chuv.ch (A.V.); pascal.monnin@chuv.ch (P.M.); 3Institute of Radiation Physics, Lausanne University Hospital (CHUV), CH 1007 Lausanne, Switzerland; 4CT/AMI Research and Development, Philips Medical Systems, Haifa 31004, Israel; elias.lahoud@philips.com (E.L.); yoad.yagil@philips.com (Y.Y.); 5Philips GmbH Innovative Technologies, Philips Research Laboratories, 22335 Hamburg, Germany; klaus.erhard@philips.com; 6Radiology Department, Hospices Civils de Lyon, 69500 Lyon, France; salim.si-mohamed@chu-lyon.fr (S.A.S.-M.); loic.boussel@chu-lyon.fr (L.B.); philippe.douek@chu-lyon.fr (P.C.D.); 7Faculté de Médecine Lyon Est, Université Claude Bernard Lyon 1, CREATIS, CNRS UMR 5220, INSERM U1206, INSA-Lyon, 69100 Lyon, France; 8Department of Medical Imaging, CHU Nimes, University of Montpellier, 30900 Nimes, France; joel.greffier@chu-nimes.fr

**Keywords:** computed tomography angiography, coronary vessels, cardiac imaging techniques, phantoms imaging, image quality enhancement

## Abstract

Aims: To evaluate spectral photon-counting CT’s (SPCCT) objective image quality characteristics in vitro, compared with standard-of-care energy-integrating-detector (EID) CT. Methods: We scanned a thorax phantom with a coronary artery module at 10 mGy on a prototype SPCCT and a clinical dual-layer EID-CT under various conditions of simulated patient size (small, medium, and large). We used filtered back-projection with a soft-tissue kernel. We assessed noise and contrast-dependent spatial resolution with noise power spectra (NPS) and target transfer functions (TTF), respectively. Detectability indices (d’) of simulated non-calcified and lipid-rich atherosclerotic plaques were computed using the non-pre-whitening with eye filter model observer. Results: SPCCT provided lower noise magnitude (9–38% lower NPS amplitude) and higher noise frequency peaks (sharper noise texture). Furthermore, SPCCT provided consistently higher spatial resolution (30–33% better TTF_10_). In the detectability analysis, SPCCT outperformed EID-CT in all investigated conditions, providing superior d’. SPCCT reached almost perfect detectability (AUC ≈ 95%) for simulated 0.5-mm-thick non-calcified plaques (for large-sized patients), whereas EID-CT had lower d’ (AUC ≈ 75%). For lipid-rich atherosclerotic plaques, SPCCT achieved 85% AUC vs. 77.5% with EID-CT. Conclusions: SPCCT outperformed EID-CT in detecting simulated coronary atherosclerosis and might enhance diagnostic accuracy by providing lower noise magnitude, markedly improved spatial resolution, and superior lipid core detectability.

## 1. Introduction

Since its inception in 1973, computed tomography (CT) has undergone steady improvements on both the data acquisition and image reconstruction aspects, quickly becoming a key player in cardiovascular imaging and establishing itself as the primary non-invasive coronary artery disease assessment tool [1]. Although the initial CT report by Hounsfield already mentioned the potential advantages of acquiring data at various energy levels [2], CT systems capable of collecting two distinct energy bands routinely were made available commercially only three decades later, with the introduction of dual-energy CT (DECT) platforms. DECT is becoming widely available clinically and can now be used to improve patient safety and diagnostic performance in everyday practice. In cardiovascular medicine, in particular, DECT helps reduce iodine dose [3,4], improve vessel opacification [5], save radiation dose with virtual non-contrast reconstruction [6], among others. On the other hand, DECT suffers from some fundamental limitations, including the absence of notable improvement in spatial resolution or electronic noise compared with single-energy systems, which could be addressed by photon-counting-detector (PCD) technology [7].

PCDs’ principle is to operate without generating visible light inside detector elements, thereby eliminating the challenges related to scintillators and associated electronic noise while providing a refined spectral analysis. Contrary to conventional energy-integrating detectors (EID)—which are used in single-energy and DECT systems—that measure the total energy deposited in the detector, PCDs quantify the energy of each incident photon according to two or more thresholds called “energy bins” and can be produced with a much smaller detector element size to increase spatial resolution. For these reasons, PCD-CT is expected to address some major limitations of EID-based DECT [8,9,10].

Coronary CTA (CCTA) is one of the most demanding CT imaging examinations due to heart motion and high spatial resolution requirements [11]; it is especially challenging to perform because the gantry needs to be operated at maximum speed for the sake of temporal resolution. Because CCTA requires both low noise and great anatomical detail, it remains a challenging examination, especially in overweight subjects. Technological advances constantly push the limits of the possible and promote CCTA as a reproducible, accurate, and reliable diagnostic test. The advent of PCD-CT is one of these technological advances that can shift the patient management paradigm in the next ten years. EIDs have been in use for almost five decades and have been CT’s backbone until now, more recently bringing CCTA to clinical routine.

Our purpose was to thoroughly characterize the image quality properties of a preclinical spectral photon-counting CT (SPCCT) prototype compared with a clinical standard-of-care EID-CT system in the setting of CCTA. To this end, we measured image noise and contrast-dependent spatial resolution properties under various simulated patient size conditions. Joint effects of noise properties and spatial resolution were modeled using state-of-the-art mathematical model observers, evaluating the systems’ performance to detect simulated non-calcified atherosclerotic plaques and lipid core in CCTA.

## 2. Materials and Methods

### 2.1. Experimental Design

We used a custom-made (Lausanne University Hospital, Lausanne, Switzerland) 10-cm-diameter cylindrical module made of low-density polyethylene (PE, average CT number at 100 kVp ≈ −100 HU). This module had a 5-cm-diameter central hole that was filled with an iodinated contrast material (CM) solution (Iomeprol 400 mixed with normal saline; Iomeron 400^®^, Bracco Imaging France, Massy, France) at a concentration of 18 mg I/mL, yielding CT numbers in the range of clinical CCTA at 100 kVp (approximately 350 HU) [4]. This CM solution and the surrounding PE created a pair of materials approaching the object-to-background contrast difference (|ΔHU|) encountered in CCTA, assuming that coronary arteries are opacified by iodinated CM and are surrounded by epicardial fat whose CT number is around −100 HU. The transition between PE and the CM solution served to measure the contrast-dependent spatial resolution using the target transfer function (TTF), an advanced physical metric particularly suited for CT taking into account the contrast-dependency of spatial resolution [12,13]. We inserted the CCTA module into an anthropomorphic thorax phantom (QRM, Moehrendorf, Germany) that was scanned as is (“small” patient size) and with additional fat-mimicking extension rings to simulate heavier bodyweights (“medium” and “large” patient size). The corresponding approximate patient weights are 50 kg (small), 80 kg (medium), and 100 kg (large size). Pictures of the phantom setup are provided in Figure 1. The phantom’s background was used to compute the noise power spectrum (NPS), a further advanced image quality metric providing a comprehensive assessment of noise by plotting noise magnitude as a function of spatial frequency [12,13]. Approval of the institutional ethics committee was not required since no living beings were involved.

### 2.2. Acquisition Protocol and Image Reconstruction

We scanned the phantom on a clinically available 64-detector row dual-layer detector EID-CT system (IQon Spectral CT, Philips Healthcare, Haifa, Israel) following the standard clinical acquisition protocol for CCTA in our University hospital, at a dose of 10 mGy. Volume CT dose indices (CTDIvol) were computed for a 32-cm-diameter (polymethyl methacrylate) reference phantom and retrieved from radiation-dose structured reports. Dose modulation was disabled to achieve a comparable dose on both CT systems. Next, we scanned the same phantom setup using a similar acquisition protocol on a prototype SPCCT system (SPCCT, Philips Healthcare, Haifa, Israel), aiming to generate comparable datasets by matching tube potential and loading and image reconstruction parameters with the EID-CT platform. The SPCCT is a large field-of-view (500 mm) system equipped with 2 mm thick Cadmium-Zinc-Telluride detectors yielding a pixel pitch of 270 × 270 µm at isocenter and a z-coverage of 17.5 mm arranged in 64 detector rows. Each detector channel has its own application-specific integrated circuit providing discrimination of five separate energy bands. Further technical details can be found here [14,15]. To ensure optimal precision of image quality metrics, the phantom was scanned eight times consecutively on each CT system, without any repositioning or parameter variation, to obtain datasets with a sufficiently large number of images. These eight acquisitions served to improve the statistics and are not intended to evaluate measurement uncertainties. Evaluating variation and uncertainty when using model observers is challenging and currently lacking a proven methodology. Table 1 presents the detailed settings for data acquisition and image reconstruction.

Images were reconstructed using the “high-resolution B” kernel (a similar one for both CT systems), which is suited for coronary artery imaging. Detector-based spectral CT system provided true conventional reconstruction; only the latter were analyzed as part of this research. Additionally, to keep the comparison as accurate and fair as possible, we refrained from using advanced image reconstruction algorithms such as iterative reconstruction. The goal of this experiment is to understand the detectors’ raw performance. This resulted in a total of 6 different CT datasets available for analysis: two CT systems (EID vs. SPCCT) × three phantom sizes (small, medium, and large).

### 2.3. Image Analysis

#### 2.3.1. Noise Power Spectrum (NPS)

We assessed image noise in the homogeneous background area of the phantom made of PE. We calculated noise power spectra (NPS) following the International Commission on Radiation Units and Measurements’ reports 54 and 87 [13] to quantify and characterize noise. NPS has established itself as state of the art for CT noise characterization, based on its unique ability to provide noise magnitude evaluation and noise texture analysis [16,17]. Four square regions of interest (ROI) of 100 × 100 pixels positioned at different locations in the background of the phantom (Figure 2) in 214 axial CT slices were used to compute 2D NPS (total of 856 ROIs), which were then radially averaged to yield 1D NPS. NPS was analyzed in terms of NPS peak frequency shift and noise magnitude reduction. NPS peak frequency was defined as the NPS’s maximum amplitude. The noise magnitude was defined as the integral of the area under the NPS curve.

The NPS peak frequency shift was calculated according to the following formula:(1)$${\mathrm{NPS}\;\mathrm{peak}\;\mathrm{frequency}\;\mathrm{shift}}_{i}=\frac{f_{\max\left({\mathrm{patient}\;\mathrm{size}}_{i}\;\mathrm{PCD}\right)}-f_{\max\left({\mathrm{patient}\;\mathrm{size}}_{i}\mathrm{EID}\right)}}{f_{\mathrm{peak}\left({\mathrm{patient}\;\mathrm{size}}_{i}\;\mathrm{EID}\right)}}\times 100$$

The noise magnitude reduction was calculated according to the following formula:(2)$${\mathrm{Noise}\;\mathrm{magnitude}\;\mathrm{reduction}}_{i}=\frac{\int \mathrm{NPS}\;\left({\mathrm{patient}\;\mathrm{size}}_{i}\;\mathrm{PCD}\right)-\int \mathrm{NPS}\;\left({\mathrm{patient}\;\mathrm{size}}_{i},\mathrm{EID}\right)}{\int \mathrm{NPS}\;\left({\mathrm{patient}\;\mathrm{size}}_{i},\mathrm{EID}\right)}\times 100$$
where i corresponds to small, medium, or large patient size.

#### 2.3.2. Target Transfer Function (TTF)

We investigated contrast-dependent spatial resolution in the phantom region containing the iodine solution. The transition between PE and the CM solution served to measure the TTF at a contrast close to 450 HU. A total of 214 axial CT sections were used to calculate the TTF. Square ROIs of 68 × 68 mm were extracted from the CT image to obtain 2D TTFs from edge spread functions, using an angular aperture and a pitch of 15° and 10°, respectively. 1D TTFs were subsequently generated by radially averaging 2D TTFs. Spatial resolution performances of the SPCCT and EID-CT systems were compared in terms of TTF frequency shift at 50% (TTF_50_) and 10% (TTF_10_) of its value at zero frequency for the three patient sizes. TTF frequency shifts were calculated using the following formula:(3)$${\mathrm{TTF}\;\mathrm{frequency}\;\mathrm{shift}}_{j}=\frac{f_{j\left({\mathrm{patient}\;\mathrm{size}}_{i}\;\mathrm{PCD}\right)}-f_{j\left({\mathrm{patient}\;\mathrm{size}}_{i}\;\mathrm{EID}\right)}}{f_{j\left({\mathrm{patient}\;\mathrm{size}}_{i}\mathrm{EID}\right)}}\times 100$$
where j equals 10 or 50%, and i corresponds to small, medium, or large patient size.

#### 2.3.3. Non-Pre-Whitening with Eye Filter (NPWE) Model

To account for noise magnitude, noise texture, and contrast-dependent spatial resolution at the same time, we computed detectability indices (d’) using the following model:(4)$$d\text{'}=\frac{\sqrt{2\pi }\left|\Delta \mathrm{HU}\right|\int_{0}^{f_{\mathrm{Ny}}}S^{2}\left(f\right){\mathrm{TTF}}^{2}\left(f\right){\mathrm{VTF}}^{2}\left(f\right)\mathrm{fdf}}{\sqrt{\int_{0}^{f_{\mathrm{Ny}}}S^{2}\left(f\right){\mathrm{TTF}}^{2}\left(f\right)\mathrm{NPS}\left(f\right){\mathrm{VTF}}^{4}\left(f\right)\mathrm{fdf}}}$$
where |ΔHU| is the contrast in absolute CT numbers between an object (i.e., non-calcified atherosclerotic plaque and lipid-rich plaque, respectively) and the surrounding homogenous background (i.e., coronary lumen and lipid-poor plaque, respectively), f the radial spatial frequency, fNy the radial Nyquist frequency, S the magnitude of the Fourier transform of the input signal (here, S = r/f J1(2πrf), with r the disk radius and J1 the Bessel function of the first kind), and VTF the visual transfer function of the human eye.

The model was adjusted to simulate two distinct but clinically relevant tasks. First, we assessed a high (400 HU) object-to-background contrast to simulate a non-calcified atherosclerotic plaque in the coronary artery wall [18]. The plaque was simulated as a half-disc of varying size whose upper (semicircular) portion causes lumen narrowing, and its flat portion abuts the vessel wall. The second task was designed to assess low-attenuation (also called “lipid-rich”) plaques’ detectability. Low attenuation composition is a known determinant of atherosclerotic plaque vulnerability histologically defined as a necrotic or lipidic core measuring > 200 µm [19]. According to existing data, fibrous plaques have average CT numbers of around 60 HU. In contrast, the lipid core’s CT number is close to 30 HU, meaning that the contrast between the fibrous and lipid plaque components is about 30 HU [20,21]. Consequently, we modeled the lipid core as a circular area whose object-to-background contrast |ΔHU| is 30 HU, with a diameter ranging from 0.5 to 3 mm, in 0.5 mm steps. The NPWE model provides d’ varying from 0 to infinity and is directly related to the accuracy. The link between d’ and the area under the receiver operating characteristic curve (AUC) can be used to assess the accuracy obtained for a specific task; a d’ > 2 corresponds to an AUC of 90% [12].

## 3. Results

### 3.1. Noise Power Spectrum

The prototype SPCCT impacted both the overall noise magnitude and noise texture (NPS peak frequency) compared with the clinical EID-CT, as demonstrated in Figure 3. Not only did the PCD system exhibit consistently lower noise magnitude, but it also had higher frequency noise since NPS peak frequency shifted towards high frequencies (from 0.38 to 0.47 mm^−1^) compared with the EID system (from 0.27 to 0.3 mm^−1^). This is mainly due to the fact that SPCCT has improved detection efficiency, can virtually eliminate electronic noise, and additionally, individual detector elements are manufactured in much smaller physical size, shifting the noise texture towards higher spatial frequencies. The noise component close to the zero frequency appears high on the PCD system; near-zero noise represents large-scale background inhomogeneity caused by scattered radiation, dark current, non-uniform detector gain, or beam hardening. However, its meaning is limited in medical imaging because of the human eye’s low sensitivity to low frequencies. Increasing the phantom size resulted in a stronger noise magnitude on both CT systems, with no substantial noise texture change (no NPS peak frequency shift). The SPCCT system’s NPS peaked at a 51% higher frequency at small phantom size while providing a 9% lower noise magnitude than the EID-CT (Table 2). At medium and large phantom sizes, the NPS peak frequency was 37% and 26% higher on the SPCCT, respectively, while the noise magnitude was 33% and 38% lower, respectively.

### 3.2. Target Transfer Function

The SPCCT’s spatial resolution performance was measured using the TTF and is plotted in Figure 4. The SPCCT provided a noticeably higher spatial resolution than the EID-CT, with a 35%, 37%, and 38% better TTF_50_ and 30%, 31%, and 33% better TTF_10_, for the small, medium, and large size phantoms, respectively (Table 3). Furthermore, we found that increasing the phantom size had a limited detrimental effect on the spatial resolution of both CT systems. Still, the EID-CT was slightly more prone to resolution loss at large phantom size.

### 3.3. Non-Pre-Whitening with Eye Filter Model Observer

For both non-calcified atherosclerotic plaque and lipid core detection tasks, the SPCCT outperformed the EID-CT regardless of the plaque size. Specifically, the SPCCT provided 22–43% better d’ for non-calcified plaque detection and 21–48% better d’ for lipid core characterization, depending on plaque and phantom size (Figure 5 and Figure 6). For detecting the smallest simulated non-calcified plaque (0.5 mm), both systems reached the threshold of 90% AUC (d’ > 2) with the small and medium-sized phantom. For the large phantom, only the SPCCT achieved 90% AUC (EID-CT AUC = 75%). For characterizing the lipid core, the limit of 90% AUC in the small phantom was 1.5 and 1 mm with the EID-CT and SPCCT, respectively. In the medium phantom, the EID-CT system did not reach the limit of 90% AUC—even for the largest simulated lipid core (3.0 mm)—while the SPCCT achieved 90% AUC down to a lipid core size of around 2.0 mm. Neither system achieved 90% AUC in the large phantom and 3.0 mm lipid core, but the SPCCT system achieved 85% AUC, whereas EID-CT reached only 77.5% AUC.

Figure 7 illustrates the visual appearance of the CCTA phantom scanned on the EID-CT and SPCCT systems at varying phantom sizes. Image noise increased at medium and large sizes. Still, regardless of the simulated patient size, the iodinated solution vs. PE interface appeared sharper on the SPCCT system, confirming the trends demonstrated in the quantitative analysis.

## 4. Discussion

Our phantom study assessed the feasibility of CCTA using a prototype SPCCT system and compared the system’s performance with the current clinical standard of care that is a dual-layer EID-CT system. We characterized the image quality using NPS (image noise) and TTF (spatial resolution) metrics and performed a specific task-based investigation of CCTA. We showed that at equivalent regular radiation dose (10 mGy), the SPCCT operating with PCDs provides solid performance for detecting non-calcified atherosclerotic plaque and lipid-rich components down to a size of 0.5 mm and 1.5 mm, actually outperforming the EID-CT system. Cardiac PCD-CT has recently been used in an animal study advocating the transition from EID to PCD [22], and our work is a step further in that direction.

NPS analysis confirmed the significantly lower noise magnitude of SPCCT, which is an anticipated improvement [23] owing to the ability of PCDs to void electronic noise almost completely and showed differences in noise texture, with NPS peaks occurring at a significantly higher spatial frequency with the SPCCT. Higher frequency peak visually translates to “finer” noise texture, facilitating lesion detection [17], especially when small and with low object-to-background contrast, such as the lipid-rich plaques we simulated in the frequency domain with our task-based assessment. A further noteworthy fact was that the noise increase associated with larger phantom sizes was steeper on the EID-CT, which bodes well for dose savings while maintaining appropriate image quality in overweight patients with PCD-CT in the future.

The systems’ spatial resolution performance assessment also showed considerable differences. Of note, the SPCCT provided markedly improved spatial resolution over the conventional CT system, on par with recent previous investigations [23,24,25]. Because they do not require septa physically separating detector elements, PCDs can be manufactured in much smaller dimensions, in the order of 100–500 µm, overcoming one of EIDs’ major limitations: dose-inefficiency at small detector element size. In PCDs, pixels are recovered by anode parceling and can be subdivided if needed. Interestingly, our study showed that the SPCCT is also less prone to spatial resolution deterioration when scanning the large phantom. Spatial resolution is critically important in CCTA for three main reasons: first to resolve small atherosclerotic plaques in any plane since coronary arteries measure ≤ 2 mm distally, meaning that as little as 4–6 voxels may be available to quantify stenoses depending on the intrinsic spatial resolution [26]. PCD can utilize sharper filters to enable more voxels, with higher noise that can be managed with noise reduction algorithms. Second, the detection of lipid-rich plaque components has a predictive value [19] but is challenging due to the lipid-rich core’s small size [21,27], hence the need for higher spatial resolution. Our model observer assessment demonstrated SPCCT’s potential to address both of these issues, providing consistently higher d’ than conventional EID-CT. Model observers are particularly relevant image quality indicators because they exhibit a stronger correlation with human observer performance than the classic contrast-to-noise ratio [28]. On the other hand, model observers do not allow for an anatomic representation of the image’s features because lesions are simulated in the frequency domain. The third reason CCTA requires high spatial resolution is that, in calcified plaques, blooming artifacts can lead to stenosis overestimation [29]. This is caused by the convolution of the system’s point-spread function [30], and PCD-CT has shown promising results in mitigating this effect [24,31]. Additionally, SPCCT can improve vascular imaging in the presence of metal stents that cause blooming artifacts for the same reasons as calcium [32,33].

Our study has several limitations. First, we demonstrated that SPCCT yields higher spatial resolution, even though we reconstructed only 512^2^ matrices. The underlying reasons were first to keep the investigation clinical since 512^2^ matrices are standard-of-care, and second, we wanted to keep noise low enough, especially with the large phantom. SPCCT offers larger matrices, including 1024^2^ or 2048^2^, with sharper reconstruction filters that could potentially be clinically relevant for stenosis quantification, particularly in calcified plaques with associated blooming artifacts. Larger matrices come at the cost of increased noise but could be combined with advanced reconstruction algorithms in the future, such as iterative or deep-learning-based reconstructions [34,35]. Furthermore, this study did not assess calcified plaque, which challenges CCTA interpretation when present in high quantities. As mentioned earlier, PCD-CT has already shown promise to improve calcified coronary artery analysis ex vivo [24] and appears to yield more accurate [36] and reduced-radiation dose [37] coronary calcium scoring. Additionally, by design, we investigated only one radiation dose level because we aimed to understand the effect of patient size. While we could have used higher doses for the large phantom, the future trend will be to reduce the radiation dose for normal-sized patients instead of increasing the dose for large patients. Although the NPWE model observer has been extensively validated against human performance [13,38], a study design evaluating human performance specifically for CCTA would consolidate our conclusions. Finally, the cardiac phantom was static, which may lead to an overestimation of the detectability performance. However, at equal gantry revolution time, the overestimation magnitude is the same for both systems; noise and spatial resolution properties are given by the systems’ design.

Future directions might include further advanced experiments since various innovative approaches are being developed. 3D TTF is an example [39] that would provide an estimation of spatial resolution not only in-plane but also in the z-direction; the latter is relevant because it may differ from the TTF measured in the x-y plane, especially in a non-linear environment such as when using iterative reconstruction. However, specific methods need to be developed to achieve 3D TTF analysis. As mentioned earlier, the evaluation of model observers’ uncertainties and the reproducibility of CT system’s performance are currently lacking and would constitute an area of original research to derive and validate a mathematically sound method. One possible approach could be to carry out comparisons between different departments or research groups using d’ in order to measure variations. Furthermore, a more complete understanding of the relationship between the numbers of scans used for each approach and the resulting uncertainty in the performance metric is an area of active investigation, as discussed in the American Association of Physicists in Medicine (AAPM) report no. 233. Future research addressing the potential of dose reduction with iterative reconstruction is needed to manage patients better because filtered-back projection is no longer a clinical standard. Potential dose reduction can be expected from the d’ charts presented in our study. However, care must be taken when comparing advanced image quality metrics (NPS, TTF, d’) while using non-linear techniques such as iterative reconstructions. The latter affect both noise texture and spatial resolution [17], thus influencing d’ beyond the expected impact of detector performance. Finally, research based on innovative phantom designs including ribs, textured backgrounds, vessels inserts, and calcified plaque, as well as 3D-printed inserts representing the coronary artery tree [40], may further support the shift from EID to PCD systems.

## 5. Conclusions

SPCCT outperformed conventional EID-CT in the task of detecting simulated non-calcified and lipid-rich plaque in coronary arteries, more so with the large phantom. The SPCCT’s lower noise and higher spatial resolution could be translated into improved accuracy for stenosis quantification and plaque characterization or reduced radiation dose, particularly in large patients often subjected to increased radiation dose and decreased diagnostic performance tests.

## Figures and Tables

**Figure 1 diagnostics-11-02376-f001:**
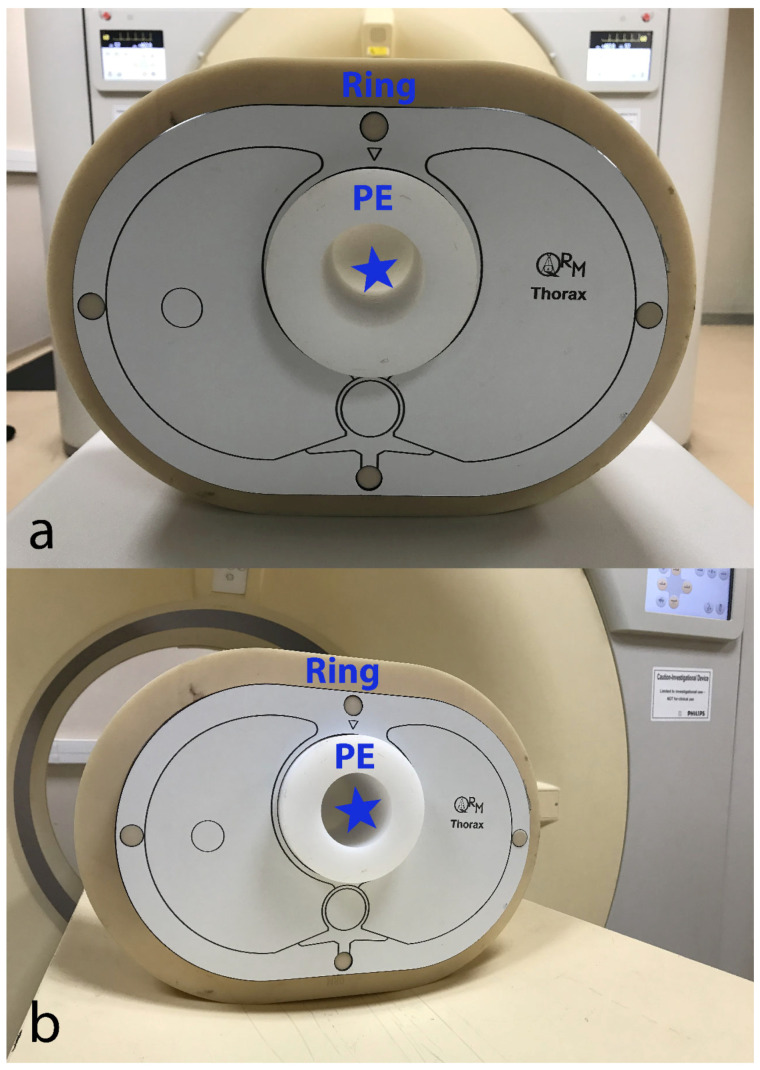
EID-CT (**a**) and SPCCT (**b**) systems with the phantom setup that was placed at the isocenter. CCTA module made of PE is shown with an empty cavity (blue star) to be filled with iodinated contrast material solution for the experiments. The anthropomorphic thorax phantom is shown with a fat-mimicking extension ring (“medium” patient size configuration). CCTA—coronary computed tomography angiography; EID-CT—energy-integrating detector computed tomography; PE—polyethylene; SPCCT—spectral photon-counting computed tomography.

**Figure 2 diagnostics-11-02376-f002:**
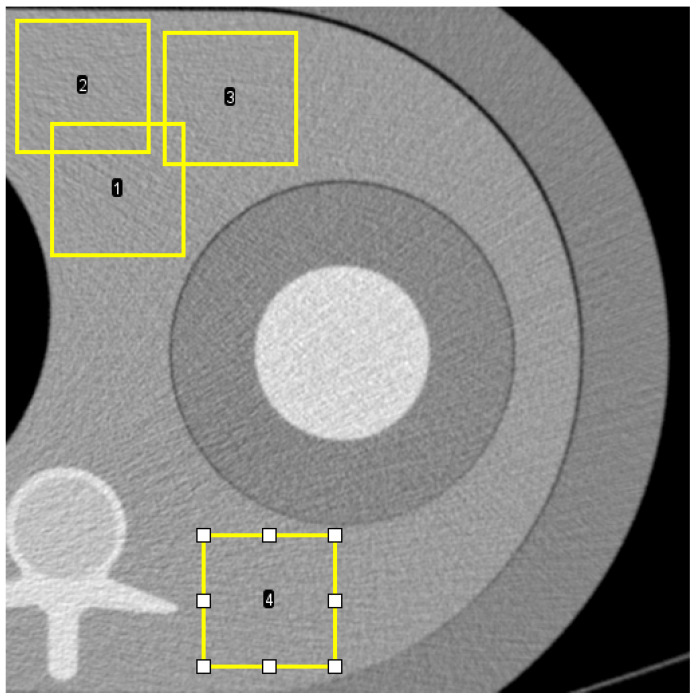
Axial CT image shows four examples (1, 2, 3, and 4) of region-of-interest (ROI) placement in the medium phantom for calculation of noise power spectrum (NPS). The position is similar in small and large phantoms.

**Figure 3 diagnostics-11-02376-f003:**
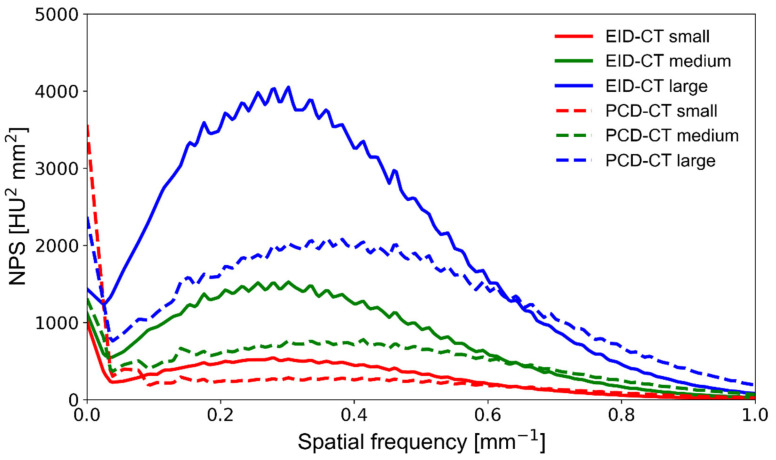
NPS curves obtained on a clinical EID-CT (solid lines) and a prototype PCD-CT (dashed lines) system at various phantom sizes. The area under the curve is representative of the noise magnitude, whereas the NPS center frequency indicates differences in noise texture. NPS—noise power spectrum; PCD-CT—photon-counting detector computed tomography; EID-CT—energy-integrating detector computed tomography.

**Figure 4 diagnostics-11-02376-f004:**
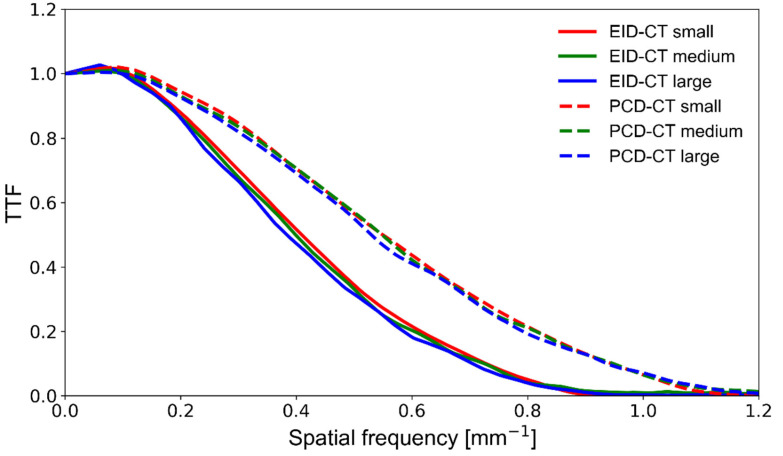
TTF curves obtained on a clinical EID-CT (solid lines) and a prototype PCD-CT (dashed lines) system at various phantom sizes. The area under the curve indicates spatial resolution performance. TTF—target transfer function; PCD-CT—photon-counting detector computed tomography; EID-CT—energy-integrating detector computed tomography.

**Figure 5 diagnostics-11-02376-f005:**
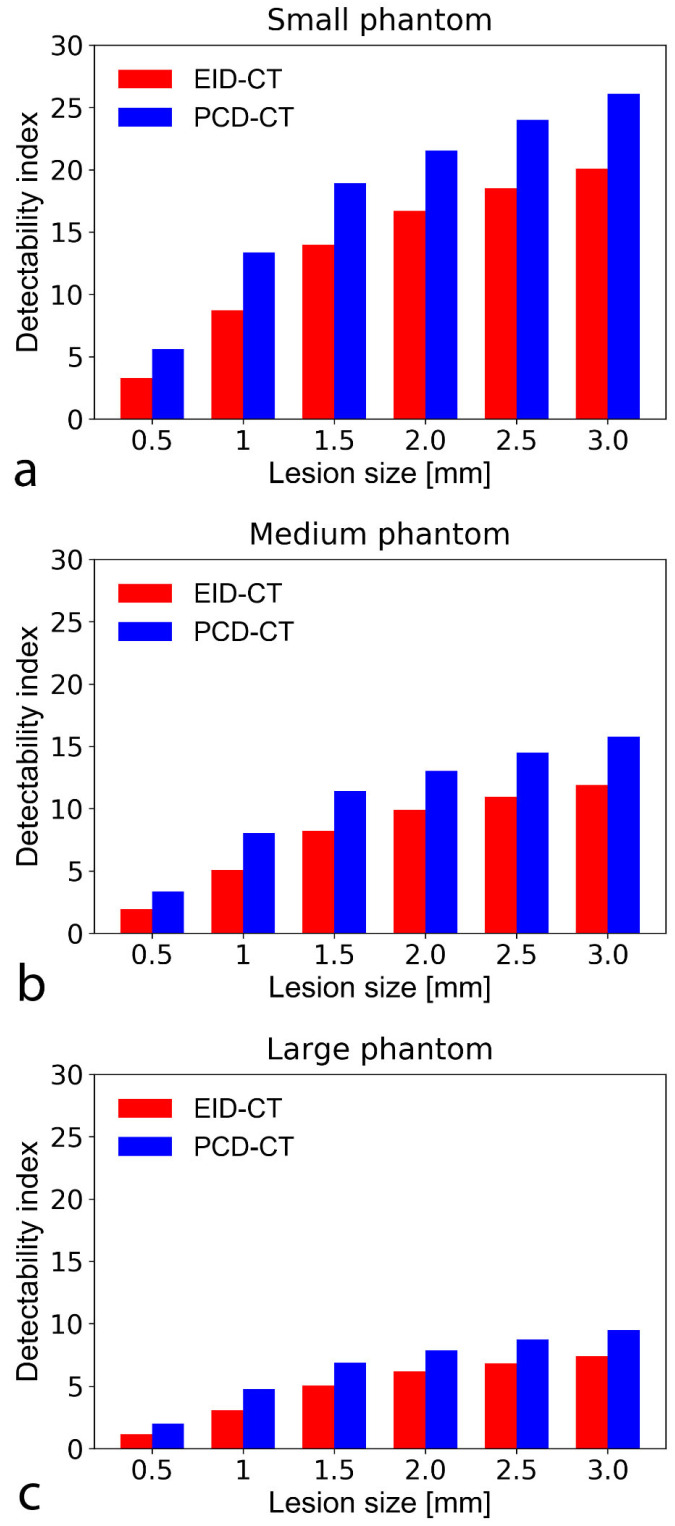
Bar chart show detectability indices (d’) of non-calcified atherosclerotic plaque with an object-to-background contrast |ΔHU| of 450 HU and CTDI = 10 mGy in the small (**a**), medium (**b**), and large sized (**c**) phantom setup. A d’ of 2 corresponds to 90% accuracy (AUC). The SPCCT consistently provided higher detectability indices than the conventional system. Note that at large phantom size, only the PCD-CT system could accurately detect (i.e., with a d’ ≥ 2 indicating an AUC of 90%) the smallest simulated plaque (0.5 mm). CTDI—computed tomography dose index; PCD-CT—photon-counting detector computed tomography. CTDI—computed tomography dose index; EID-CT—energy-integrating detector computed tomography; AUC—area under the curve.

**Figure 6 diagnostics-11-02376-f006:**
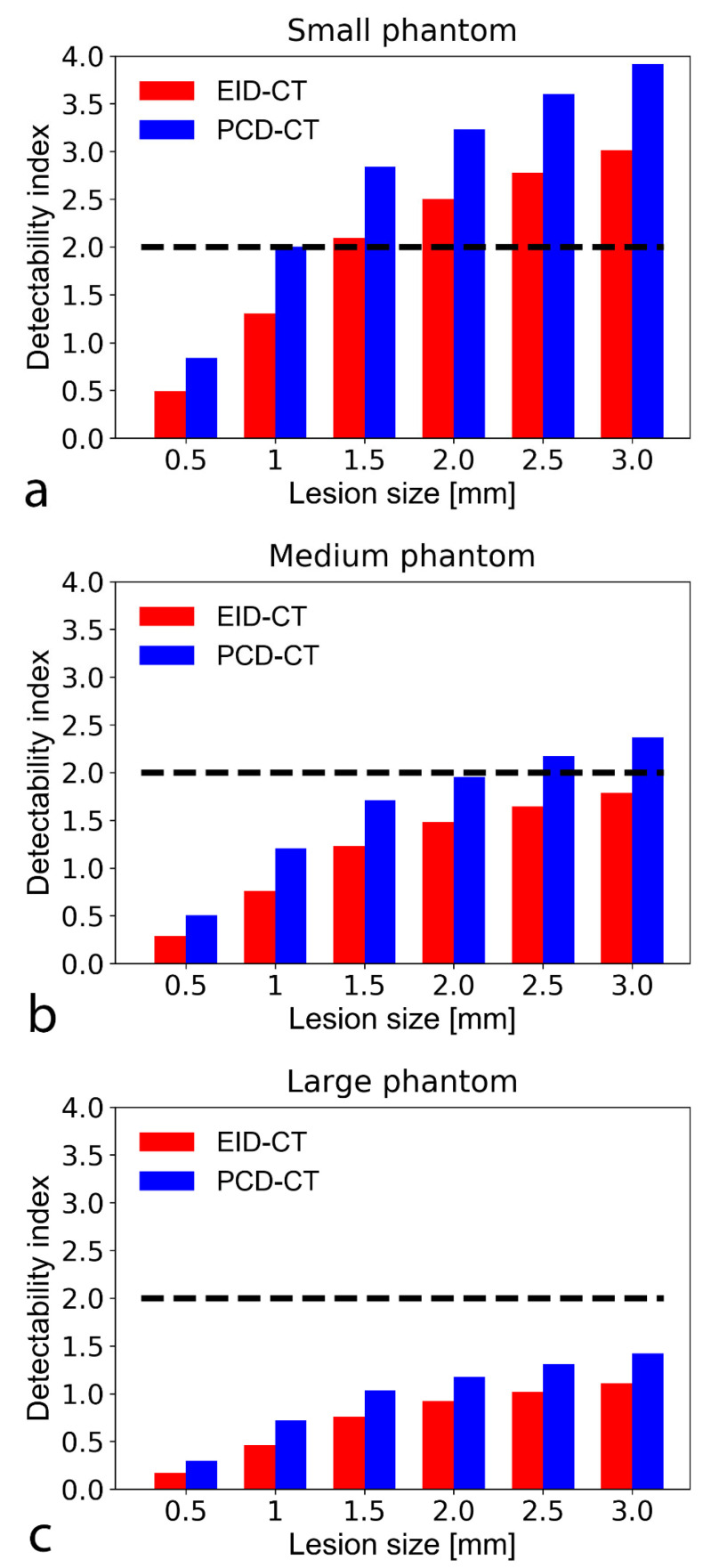
Bar chart shows detectability indices (d’) of lipid-rich atherosclerotic plaque with an object-to-background contrast |ΔHU| of 30 HU in the small (**a**), medium (**b**), and large sized (**c**) phantom setup. A d’ of 2 corresponds to 90% accuracy (AUC), plotted on the graphs as a black dashed line. The PCD-CT consistently provided higher detectability indices than the conventional system. At the tested CTDI of 10 mGy, neither the EID nor the SPCCT reached 90% AUC to detect a 0.5 mm lipid core. With the small phantom, the EID and SPCCT systems reached 90% AUC down to a lipid core size of 1.5 and 1 mm, respectively. AUC—area under the curve; CTDI—computed tomography dose index; EID-CT—energy-integrating detector computed tomography; PCD-CT—photon-counting detector computed tomography.

**Figure 7 diagnostics-11-02376-f007:**
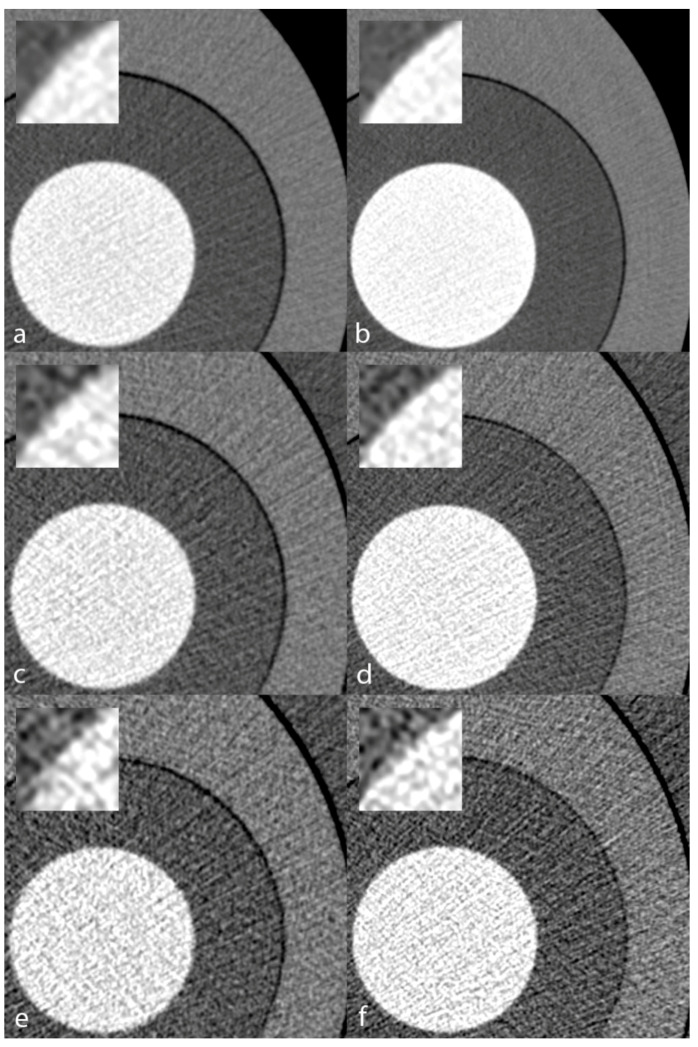
Visual appearance of the TTF phantom inserted in a small (**a**,**b**), medium (**c**,**d**), and large (**e**,**f**) anthropomorphic chest phantom. Conventional reconstructions obtained from acquisitions on the EID-CT (**a**,**c**,**e**) and the PCD-CT (**b**,**d**,**f**) systems. Zoomed views of the polyethylene/iodinated solution transition better depict the finer noise texture and sharper transition yielded by the PCD-CT. TTF—target transfer function; PCD-CT—photon-counting detector computed tomography; EID-CT—energy-integrating detector computed tomography.

**Table 1 diagnostics-11-02376-t001:** Detailed settings for data acquisition and image reconstruction for the investigated CT angiography protocols on both CT systems.

CT System	EID-CT	PCD-CT
**Radiation dose level**		
CTDIvol (mGy)	10	10
**Data acquisition**		
Tube potential (kVp)	120	120
Tube current (mA)	330	330
Gantry revolution time (s)	0.5	0.5
Beam collimation (mm)	32 × 0.672	64 × 0.2724
Scan mode	Helical	Helical
Automatic exposure control	Off	Off
**Image reconstruction**		
Display field of view (mm)	200 × 200	200 × 200
Matrix size	512 × 512	512 × 512
Section thickness (mm)	0.6	0.6
Section increment (mm)	0.6	0.6
Kernel	High-res B	PCD-High-res B
Algorithm	Filtered back-projection	Filtered back-projection

CTDIvol—volume CT dose index, EID—energy-integrating detector, PCD—photon-counting detector.

**Table 2 diagnostics-11-02376-t002:** Noise magnitude reduction and NPS peak frequency shift in percentage differences for PCD-CT in comparison with EID-CT at the three investigated patient sizes.

Phantom Size	Noise Magnitude Reduction (%)	Peak Frequency Shift (%)
Small	−9	47
Medium	−33	37
Large	−38	27

NPS—noise power spectrum; PCD-CT—photon-counting computed detector computed tomography; EID-CT—energy-integrating computed tomography.

**Table 3 diagnostics-11-02376-t003:** TTF frequency shifts (percentage differences) for PCD-CT compared with EID-CT at the three investigated patient sizes.

	TTF Frequency Shifts (%)	
Phantom Size	TTF_50_	TTF_10_
Small	35	30
Medium	37	31
Large	38	33

TTF—target transfer function; PCD-CT—photon-counting computed detector computed tomography; EID-CT—energy-integrating computed tomography.

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
