# Peer review of "Performance of Spectral Photon-Counting Coronary CT Angiography and Comparison with Energy-Integrating-Detector CT: Objective Assessment with Model Observer"

_diagnostics, 2021, doi:10.3390/diagnostics11122376_

Round 1

Reviewer 1 Report

This paper is timely as photon counting is exploding in particular in CT, especially in the past year. The authors have performed measurements based on classical image quality metrics in and anthropomorphic phantom, considering three patient sizes.

I have some major concerns about some of the methods used and in my opinion, the study needs to be extended, and include more experiments.

  1. The authors have only compared the performance of the PC system and a clinical standard CT system, only considering FBP reconstruction, which is not used currently in the standard clinic. I understand that this was a first step to compare basically the detectors performance, but I think that the inclusion of results with a more up to data reconstruction method in Philips systems,  with the iterative reconstruction level setup at a relevant setting compared to clinical coronary angiography, should be added.

2. The use of the NPWE model observer applied on simulated lesions in the frequency domain, has limited use in my opinion in this case. I think it would be of far more value to assess the detectability of real lesion surrogates in and anthorpomorphic phantom, and the authors have not aknowledge the limitation of their approach to assess PC technology. Though I know MOs are used on simulated lesions, I believe it would make far more stronger point in the imaged objects in PC and EID CT system were real.

3. The image quality metrics used, such as NPS and TTF were only analyzed in 2D. I suggest to extend these metrics to 3D to compare both CT systems.

4. There is a total lack of errors and uncertainty evaluation through all the paper that needs to be corrected. TTF needs to have some errors associated, as well as d's. On page 4, line 140, the authors say that "These eight acquisitions served to improve the statistics and are not intended to evaluate measurement uncertainties". In my opinion, this is not a robust justification, all the measurements provided need to have errors and define an appropriate way to assess the performance and repeatability of PC and EID CT systems performance.

Minor comments

5. Table 1. Please add if AEC was activated for this specific protocol. Please also add information in the table or text about the differences (or lack of) in filtration between both CT systems.

6. Pg 4, line 148. The justification of not using IR, is not strong enough. FBP is a starting point of comparison in performance, but as it is not used in the clinic, I strongly recommend to extend the paper including the results for IR, with whichever reconstruction filter and level of IR recommended by the manufacturer for coronary angiography.

7. Pg 4, line 159. Where were the ROIs for NPS measured? On the PE ring? On the lung inserts? Please add a CT image of the full phantom with the exact locations of the ROIs.

8. You have performed your study for the three patient sizes at 120kV, but if your biggest phantom were a real patient, would they still be scanned at 120kV or at a higher kV in the clinic? Please comment on this.

9. Please address error estimation for all your image quality metrics.

10. I think one of the promises of PC systems is potential dose reduction. This study was based on the clinical dose level that would be used I suppose with a standard size patient (please comment on this on discussion too, linked to my comment 8). Please address, at least as future work, to compare the performance of PC and EID at lower energy settings, especially in clinical practice with IR, the NPS, TTF... would be affected and potentially the performance of PC and EID would differ not only in magnitude but other effects appear.

11. Pg 11, line 328. About the model observer results, please emphasize clearly that the lesions were simulated in the frequency domain, no actual comparison with human observers has been performed in this case. As stated in comment 4, I believe the model observer is of limited value to compare both systems, in a more realistic clinical setup with IR, and a phantom with real lesions would be preferable.

12. Please add a paragraph commenting further on phantom limitations, such as the lack of tissue texture, ribs, and also realistic depiction of arteries/veins. Some commercial phantoms also offer the option of movement, such as (https://bmcmedimaging.biomedcentral.com/articles/10.1186/s12880-021-00680-7) and the inclusion of cylinder as calcification surrogates. Some other static options are also available (https://www.qrm.de/en/products/cardiac-calcification-phantom/). 3D printed anthropomorphic phantoms can also be a potential option to create surrogate inserts with realistic lesions and vessels. Please comment on this issues in Discussion with a brief overview of state of the art phantoms that could be potentially used for coronary angiography CT.

13. The used phantom lacks any tissue texture, several approaches are currently explored, mostly with 3D printing to solve this, including plastics and paper and doped ink combinations. Please comment on how more realistic phantoms could help to characterize the PC CT systems and others closer to the actual clinical image quality with patients. Please add some relevant references on this regard.

14. Please add a paragraph about your future work, new measurements, other image quality metrics to address, comparison of CT systems at other dose levels...

15. Please make regular searches during the review process to include any relevant recently published abstract or paper related to your study, as PC is a very hot topic at the moment.

Author Response

This paper is timely as photon counting is exploding in particular in CT, especially in the past year. The authors have performed measurements based on classical image quality metrics in and anthropomorphic phantom, considering three patient sizes.

I have some major concerns about some of the methods used and in my opinion, the study needs to be extended, and include more experiments.

The authors have only compared the performance of the PC system and a clinical standard CT system, only considering FBP reconstruction, which is not used currently in the standard clinic. I understand that this was a first step to compare basically the detectors performance, but I think that the inclusion of results with a more up to data reconstruction method in Philips systems, with the iterative reconstruction level setup at a relevant setting compared to clinical coronary angiography, should be added.
Response: thank you for the time spent reviewing our manuscript. Indeed, FBP has mostly disappeared from clinical protocols, however, the two systems have fundamental physical differences, and in order to mitigate biases from reconstruction protocols, we prioritized protocol similarity. Iterative reconstruction algorithms are not the same on both systems and may bias our understanding of raw detector performance (we revised the methods to address this). Evaluating iterative and potentially other reconstruction types will constitute future works as mentioned in the limitations.

The use of the NPWE model observer applied on simulated lesions in the frequency domain, has limited use in my opinion in this case. I think it would be of far more value to assess the detectability of real lesion surrogates in and anthorpomorphic phantom, and the authors have not aknowledge the limitation of their approach to assess PC technology. Though I know MOs are used on simulated lesions, I believe it would make far more stronger point in the imaged objects in PC and EID CT system were real.
Response: the reviewer is correct; NPWE performs simulations in the frequency domain. Not only does it simulate a lesion of a particular contrast and shape, but it does simulate a task and is corrected for the humans’ visual response. Additionally, NPWE relies on a real contrast difference specifically tailored for CT angiography, thereby taking into account noise and contrast-dependency of spatial resolution, i.e., contrast is not simulated. Furthermore, NPWE have been extensively validated against human performance (refs. 38-39) and this method has already been used to assess PC technology: [Rajagopal JR, Sahbaee P, Farhadi F, Solomon JB, Ramirez-Giraldo JC, Pritchard WF, Wood BJ, Jones EC, Samei E. A Clinically Driven Task-Based Comparison of Photon Counting and Conventional Energy Integrating CT for Soft Tissue, Vascular, and High-Resolution Tasks. IEEE Trans Radiat Plasma Med Sci. 2021 Jul;5(4):588-595]. We mentioned in the limitations that future experiments with human observer assessment would add relevant evidence.

The image quality metrics used, such as NPS and TTF were only analyzed in 2D. I suggest to extend these metrics to 3D to compare both CT systems.
Response: we thank the reviewer for this insightful suggestion. Indeed most if not all task-based assessments of photon-counting CT currently published are based on the well-validated 2D method: [Rajagopal JR, Sahbaee P, Farhadi F, Solomon JB, Ramirez-Giraldo JC, Pritchard WF, Wood BJ, Jones EC, Samei E. A Clinically Driven Task-Based Comparison of Photon Counting and Conventional Energy Integrating CT for Soft Tissue, Vascular, and High-Resolution Tasks. IEEE Trans Radiat Plasma Med Sci. 2021 Jul;5(4):588-595]; [Si-Mohamed SA, Greffier J, Miailhes J, Boccalini S, Rodesch PA, Vuillod A, van der Werf N, Dabli D, Racine D, Rotzinger D, Becce F, Yagil Y, Coulon P, Vlassenbroek A, Boussel L, Beregi JP, Douek P. Comparison of image quality between spectral photon-counting CT and dual-layer CT for the evaluation of lung nodules: a phantom study. Eur Radiol. 2021 Jun 29.]; [Rajagopal JR, Farhadi F, Solomon J, Sahbaee P, Saboury B, Pritchard WF, Jones EC, Samei E. Comparison of Low Dose Performance of Photon-Counting and Energy Integrating CT. Acad Radiol. 2021 Dec;28(12):1754-1760.]. While 3D TTFs are feasible, their analysis would constitute another research project as this would require specific methods to be validated. Besides, 3D analysis will be particularly interesting in a non-linear environment such as when using iterative reconstructions. We have added this relevant comment in the future directions’ paragraph.

There is a total lack of errors and uncertainty evaluation through all the paper that needs to be corrected. TTF needs to have some errors associated, as well as d's. On page 4, line 140, the authors say that "These eight acquisitions served to improve the statistics and are not intended to evaluate measurement uncertainties". In my opinion, this is not a robust justification, all the measurements provided need to have errors and define an appropriate way to assess the performance and repeatability of PC and EID CT systems performance.
Response: we appreciate the comment and acknowledge that this question has been a matter of debate in the past. Previous studies – including those mentioned above – do not usually report uncertainties because doing so is particularly challenging, and a valid method to do it is not currently available. It is, therefore, advisable not to report uncertainties rather than report biased or wrong uncertainties. Assessing the performance and repeatability of any CT system with d’ should be addressed in a full original paper specifically aiming at developing and validating a method to do so, e.g., by carrying out inter-comparisons between different departments or research groups using d’ in order to derive variations. Finally, a more complete understanding of the relationship between the numbers of scans used for each approach and the resulting uncertainty in the performance metric is an area of active investigation [AAPM report 233]. We have expanded our methods and future directions sections to better cover these critical questions.

Minor comments

Table 1. Please add if AEC was activated for this specific protocol. Please also add information in the table or text about the differences (or lack of) in filtration between both CT systems.
Response: thanks for the suggestion, we revised accordingly.

Pg 4, line 148. The justification of not using IR, is not strong enough. FBP is a starting point of comparison in performance, but as it is not used in the clinic, I strongly recommend to extend the paper including the results for IR, with whichever reconstruction filter and level of IR recommended by the manufacturer for coronary angiography.
Response: we agree with the reviewer, FBP is gradually disappearing from clinical use. For this initial study, however, a high level of priority was put on the ability to compare the systems’ characteristics and not reconstruction algorithm performance. Evaluating reconstruction algorithm performance would constitute another study because the data is unavailable. We have revised the limitations paragraph to acknowledge the fact that FBP is not used in clinical routine anymore.

Pg 4, line 159. Where were the ROIs for NPS measured? On the PE ring? On the lung inserts? Please add a CT image of the full phantom with the exact locations of the ROIs.
Response: we are grateful for the suggestion and added an image to understand better where the ROIs for the NPS were extracted.

You have performed your study for the three patient sizes at 120kV, but if your biggest phantom were a real patient, would they still be scanned at 120kV or at a higher kV in the clinic? Please comment on this.
Response: thank you for this interesting question. PCCT systems are becoming increasingly available in clinical practice however, switching tube potential is not as straightforward as with EID systems because the whole system needs a calibration procedure to be run in order to change tube current or voltage. For this reason, even larger patients would expectedly be scanned at 120 kVp, especially owing to the PCCT’s high detection efficiency and low noise characteristics.

Please address error estimation for all your image quality metrics.
Response: we appreciate the suggestion to provide error estimates for advanced image quality metrics. Current challenges are explained in point 4.

I think one of the promises of PC systems is potential dose reduction. This study was based on the clinical dose level that would be used I suppose with a standard size patient (please comment on this on discussion too, linked to my comment 8). Please address, at least as future work, to compare the performance of PC and EID at lower energy settings, especially in clinical practice with IR, the NPS, TTF... would be affected and potentially the performance of PC and EID would differ not only in magnitude but other effects appear.
Response: thanks for this constructive comment. Potential dose reduction can be expected from our d’ results. This was now added in the future directions’ paragraph.

Pg 11, line 328. About the model observer results, please emphasize clearly that the lesions were simulated in the frequency domain, no actual comparison with human observers has been performed in this case. As stated in comment 4, I believe the model observer is of limited value to compare both systems, in a more realistic clinical setup with IR, and a phantom with real lesions would be preferable.
Response: thank you for the suggestion, our statement could indeed have been misleading the reader in believing that we have evaluated images of lipid-rich plaque, but the latter were simulated in the task-based assessment. We have revised accordingly.

Please add a paragraph commenting further on phantom limitations, such as the lack of tissue texture, ribs, and also realistic depiction of arteries/veins. Some commercial phantoms also offer the option of movement, such as (https://bmcmedimaging.biomedcentral.com/articles/10.1186/s12880-021-00680-7) and the inclusion of cylinder as calcification surrogates. Some other static options are also available (https://www.qrm.de/en/products/cardiac-calcification-phantom/). 3D printed anthropomorphic phantoms can also be a potential option to create surrogate inserts with realistic lesions and vessels. Please comment on this issues in Discussion with a brief overview of state of the art phantoms that could be potentially used for coronary angiography CT.
Response: we appreciate the detailed suggestion regarding phantoms. Interesting phantom designs exist, and we have commented in the future directions paragraph. However, we want to highlight that advanced image quality metrics have been developed for the sake of totally quantitative and objective image quality analysis, which can be regarded as a strength compared to humans doing subjective image ratings on phantoms. Besides, the QRM calcification phantom has already been used for PCCT evaluation [van der Werf NR, Si-Mohamed S, Rodesch PA, van Hamersvelt RW, Greuter MJW, Boccalini S, Greffier J, Leiner T, Boussel L, Willemink MJ, Douek P. Coronary calcium scoring potential of large field-of-view spectral photon-counting CT: a phantom study. Eur Radiol. 2021 Jul 13].

The used phantom lacks any tissue texture, several approaches are currently explored, mostly with 3D printing to solve this, including plastics and paper and doped ink combinations. Please comment on how more realistic phantoms could help to characterize the PC CT systems and others closer to the actual clinical image quality with patients. Please add some relevant references on this regard.
Response: please refer to point 12.

Please add a paragraph about your future work, new measurements, other image quality metrics to address, comparison of CT systems at other dose levels...
Response: we are grateful for the suggestion and added a final paragraph regarding future works.

Please make regular searches during the review process to include any relevant recently published abstract or paper related to your study, as PC is a very hot topic at the moment.
Response: we have added recent references.

Reviewer 2 Report

1)In Fig. 2, 3 , 5, 6, the spelling needs to be corrected: PCD-CT is photon counting detector computed tomography. EID-CT is energy-integrating detector computed tomography. 

2)There seems to be a need for a future research plan on the correlation between model observer(Nonprewhitening with eye filter) and observer (radiologist) performance.

3)It would be good to insert both Spectral PC-CT system and Dual-layer EID-CT system images in fig 1.

4)It would be good to insert the selected X-ray energy band(or projection energy) in among differnt energy spectrum for FBP reconstruction algorithm in this article.

5)Do you have further performance comparison between PCD-CT system and Dual-layer EID-CT system about by using different radiation dose, image reconstruction, etc.?

Author Response

Reviewer 1

Comments and Suggestions for Authors

Thank you for the time spent to review our manuscript. Please find our responses below.

1)In Fig. 2, 3 , 5, 6, the spelling needs to be corrected: PCD-CT is photon counting detector computed tomography. EID-CT is energy-integrating detector computed tomography. 

Response: thanks for picking up this issue. We have revised the figure captions accordingly.

2)There seems to be a need for a future research plan on the correlation between model observer(Nonprewhitening with eye filter) and observer (radiologist) performance.

Response: we have added a sentence in the limitations paragraph to discuss this issue. Indeed model observers’ correlation with humans (radiologists) has been studied, but another phantom study with several humans analyzing plaques could add additional evidence.

3)It would be good to insert both Spectral PC-CT system and Dual-layer EID-CT system images in fig 1.

Response: we appreciate the suggestion and have revised Fig. 1.

4)It would be good to insert the selected X-ray energy band(or projection energy) in among differnt energy spectrum for FBP reconstruction algorithm in this article.

Response: the reviewer is right, this information was missing (we used conventional reconstructions provided by detector-based spectral CTs). We edited the Materials & Methods paragraph to provide this data.

5)Do you have further performance comparison between PCD-CT system and Dual-layer EID-CT system about by using different radiation dose, image reconstruction, etc.?

Response: this is a relevant question. By design, we aimed to address patient size at it remains a major challenge in clinical routing. We have not used iterative reconstruction because the algorithms available on both systems are not the same. Future research addressing the potential of dose reduction with iterative reconstruction is needed to better manage patient, which we added in the limitations.

Reviewer 3 Report

The authors empirically compare the performances of SPCCT with the conventional EIDCT with an appropriate setup of phantom. The manuscript has been well organized and written to show the advantages of SPCCT in several aspects. Minor suggestions are as follows. In Figure 2, the photon counting device shows much lower values of NPS than the energy integrating device case. However, the authors say inflations at relatively high frequencies. Is there any intrinsic properties that can explain such a property? In a similar manner, the photon counting devices have exceptionally high values at zero frequency. A brief explanation for the reason can be useful.

Author Response

Comments and Suggestions for Authors

The authors empirically compare the performances of SPCCT with the conventional EIDCT with an appropriate setup of phantom. The manuscript has been well organized and written to show the advantages of SPCCT in several aspects. Minor suggestions are as follows. In Figure 2, the photon counting device shows much lower values of NPS than the energy integrating device case. However, the authors say inflations at relatively high frequencies. Is there any intrinsic properties that can explain such a property? In a similar manner, the photon counting devices have exceptionally high values at zero frequency. A brief explanation for the reason can be useful.

Thank you for the time spent to review our manuscript. Please find our responses below:

  • Fig 2: SPCCT has much less area under the curve than EID-CT, but its peak spatial frequency is higher. This is mainly due the fact that SPCCT can virtually eliminate electronic noise and additionally, individual detector elements are manufactured in much smaller physical size, shifting the noise texture towards higher spatial frequencies. Additionally, the noise amplitude decreased also because of improved detection efficiency with PCD-CT.
  • Regarding the high zero frequency spatial resolution, both EID and PCD CT show more or less high values we chose not to notch. Near zero noise represents large-scale background inhomogeneity caused by scattered radiation, dark current, non-uniform detector gain, or beam hardening. However, its meaning is limited in medical imaging because this low-frequency spike does not cause a visual distortion and not have an influence in clinical routine, because of the human eye’s reduced sensitivity to low frequencies.

Reviewer 4 Report

This is a very interesting study in which the authors describe the great value of the new spectral photon counting CT (SPCCT) scanners versus the current standard scanners based on energy-integrating detection (EID-CT). They carried out multiple and well described experiments with a thorax phantom. They concluded that the SPCCT outperformed EID-CT in detecting simulated coronary atherosclerosis and it might enhance diagnostic accuracy by providing lower noise magnitude, markedly improved spatial resolution, and superior lipid core detectability. This is all substantiated by the data. For a next time, also coronary calcium detection should be included.

Author Response

Comments and Suggestions for Authors

This is a very interesting study in which the authors describe the great value of the new spectral photon counting CT (SPCCT) scanners versus the current standard scanners based on energy-integrating detection (EID-CT). They carried out multiple and well described experiments with a thorax phantom. They concluded that the SPCCT outperformed EID-CT in detecting simulated coronary atherosclerosis and it might enhance diagnostic accuracy by providing lower noise magnitude, markedly improved spatial resolution, and superior lipid core detectability. This is all substantiated by the data. For a next time, also coronary calcium detection should be included.

Response: thank you the time spent reviewing our manuscript. Indeed calcium detection is a critical aspect in coronary CTA and we shall assess calcium with another phantom design.

Round 2

Reviewer 1 Report

I appreciate the time the authors put in replying in detail to my comments. Though I still do not agree with not including iterative reconstruction, I acknowledge that the authors stated that the data is not available for new reconstructions. With the additions provided in the current manuscript, addressing the limitations, the paper can be accepted, as far as i am concerned.